# A Case Series on Genotype and Outcome of Liver Transplantation in Children with Niemann-Pick Disease Type C

**DOI:** 10.3390/children8090819

**Published:** 2021-09-17

**Authors:** Line Modin, Vicky Ng, Paul Gissen, Julian Raiman, Eva Doreen Pfister, Anibh Das, René Santer, Hanna Faghfoury, Saikat Santra, Ulrich Baumann

**Affiliations:** 1Department of Gastroenterology, Hans Christian Andersen Children’s Hospital, DK-5000 Odense, Denmark; line.modin@rsyd.dk; 2Liver Department, Birmingham Children’s Hospital, Steelhouse Ln, Birmingham B4 6NH, UK; s.santra@nhs.net; 3Division of Clinical and Metabolic Genetics, Department of Paediatrics, The Hospital for Sick Children, Toronto, ON M5G 1X8, Canada; vicky.ng@sickkids.ca (V.N.); Julian.Raiman@nhs.net (J.R.); Hanna.Faghfoury@uhn.ca (H.F.); 4NIHR Great Ormond Street Hospital Biomedical Research Centre, University College London, London WC1N 1EH, UK; p.gissen@ucl.ac.uk; 5Department of Pediatrics, Hannover Medical School, Carl-Neuberg-Str. 1, 30625 Hannover, Germany; Pfister.Eva-Doreen@mh-hannover.de (E.D.P.); das.anibh@mh-hannover.de (A.D.); 6Department of Pediatrics, "KiNDER-UKE", University Medical Center Eppendorf, Martini Str. 52 (O45), 20246 Hamburg, Germany; r.santer@uke.de; 7Division of Pediatric Gastroenterology and Hepatology, Hannover Medical School, 30625 Hannover, Germany

**Keywords:** children, liver transplantation, Niemann-Pick disease type C

## Abstract

Background: To report on clinical presentation and outcomes of children who underwent liver transplantation (LTx) and were subsequently diagnosed to have Niemann-Pick type C (NPC). Methods: Retrospective, descriptive, multi-centre review of children diagnosed with NPC who underwent LTx (2003–2018). Diagnosis was made by filipin skin test or genetic testing. Results: Nine children were identified (six centres). Neonatal acute liver failure was the most common indication for LTx (seven children). Median age at first presentation: 7 days (range: 0–37). The most prevalent presenting symptoms: jaundice (8/9), hepatosplenomegaly (8/9) and ascites (6/9). 8/9 children had a LTx before the diagnosis of NPC. Genetic testing revealed mutations in NPC1 correlating with a severe biochemical phenotype in 5 patients. All 9 children survived beyond early infancy. Seven children are still alive (median follow-up time of 9 (range: 6–13) years). Neurological symptoms developed in 4/7 (57%) patients at median 9 (range: 5–13) years following LTx. Conclusion: Early diagnosis of NPC continues to be a challenge and a definitive diagnosis is often made only after LTx. Neurological disease is not prevented in the majority of patients. Genotype does not appear to predict neurological outcome after LTx. LTx still remains controversial in NPC.

## 1. Introduction

The commonest group of known diseases causing neonatal acute liver failure (ALF) are metabolic liver disease followed by viral infections, while neonatal haemochromatosis is the single most common cause of ALF in neonates [1,2]. One of the challenges in the management of neonates with ALF is the need for rapid exclusion of contraindications for liver transplantation (LTx) before clinical deterioration of the patient. Multi-systemic disorders, such as respiratory chain defects, are ideally expeditiously diagnosed to enable decision making regarding the role of LTx [3].

Niemann-Pick disease type C (NPC) is a rare multi-systemic neurodegenerative disorder caused by a defect in intracellular lipid trafficking [4]. NPC is inherited in an autosomal recessive manner, where the majority of patients suffer prolonged and chronic neurological deterioration [5]. There is, however, a wide spectrum in reported phenotypes, both in age and mode of clinical presentation from acute to chronic [6]. In about 10% of these patients, however, the jaundice quickly worsens and leads to ALF. In children with early onset neonatal cholestasis, the clinical cause is often more aggressive and the patient usually dies before the age of 6 months [5,6]. Neonatal presentation is commonly associated with chronic liver dysfunction. This usually takes the form of a cholestatic liver disease with prolonged neonatal jaundice, often with splenomegaly [7]. Investigations for lysosomal storage disorders are often included in the diagnostic work up for neonatal cholestasis, especially when co-existent with visceral organomegaly [8]. In the absence of potent biomarkers the diagnostic process of NPC is challenging, and each diagnostic method has limitations. If screening for common mutations in the NPC1 and NPC2 genes is not fruitful, diagnosis requires the culture of fibroblasts for the detection of an abnormal lysosomal storage pattern by filipin staining and cholesterol esterification studies. However, the results are not always unambiguous. Furthermore, this is impractical as first-line screening test; however, results of alternative investigations, such as an increase of the macrophage activation marker chitotriosidase, are even less satisfactory [9,10,11]. Recent advances in biomarkers and molecular genetics have improved the diagnostic process, although definitive diagnosis remains challenging in some patients [10]. Therefore, in ALF the rapid assessment can fail to diagnose NPC before a clinical decision on LTx has to be made, although LTx is a controversial treatment strategy for patients with NPC disease, since neurological disease progression is not prevented and progressive neurological deterioration including supranuclear gaze palsy, cognitive deterioration and spasticity will appear over time. We report on clinical presentation, morbidity and mortality of a series of children who underwent LTx and were subsequently diagnosed with NPC.

## 2. Materials and Methods

We report a retrospective, descriptive case series of children diagnosed with NPC who underwent LTx between 2003 and 2018. The data is collected as a multi-centre collaboration review of clinical and laboratory features.

### 2.1. Study Design

Six centres contributed with patients: Birmingham Women’s and Children’s Hospital (UK); Children’s Health Queensland Hospital and Health Service, Department Gastroenterolgy Hepatology and Liver Transplant, Brisbane (Australia); Hannover Medical School, Clinic for Paediatric Nephrology, Hepatology and Metabolic Disorders (Germany); University Medical Center Hamburg-Eppendorf, Pediatrics, Hamburg (Germany); The Hospital for Sick Children (SickKids), Division of Gastroenterology, Hepatology and Nutrition, Toronto (Canada); and Universitario y Politecnico La Fe Valencia (Spain).

Prior to data collection, a protocol was written, a data collection sheet was developed and the study was approved according to standards of the local ethics committees. Data was collected locally in a standardised data collection sheet containing information on demographic data, symptoms at first presentation, diagnostic work-up with special focus on the NPC diagnostic work-up, data about the LTx and clinical outcome in terms of neurological symptoms at time of follow-up and mortality.

### 2.2. Patients

Clinical data from nine NPC patients were retrospectively analysed. Patients were identified by personal communication between colleagues in the field of pediatrics (*n* = 8) and a review of European liver transplant registry data (*n* = 1). Included were children in whom diagnosis was confirmed by positive filipin staining of skin fibroblasts and/or genetic testing. From peripheral blood samples, mutation analyses were performed by direct sequencing of the NPC genes where possible.

### 2.3. Statistics

All data are reported as descriptive analyses. STATA 13 (Stata Statistical Software: Release 13. College Station, TX: StataCorp LP) was used to calculate median values. All data were analysed in a descriptive way and reported as median and range and n (%) for categorical values.

## 3. Results

### 3.1. Demographic Data

Overall, a total of nine patients underwent LTx at one of six centres in Germany, UK, Canada, Spain, Australia and the USA. Eight patients came from unrelated Caucasian families, with no reported parental consanguinity, while one child was reported to have related parents. There was a gender difference in the cohort with twice as many boys (*n* = 6) compared to girls (*n* = 3).

In one family there were two affected siblings with one younger sister who has not undergone LTx. The diagnosis was made when the younger sister presented with neonatal ALF and only then was NPC diagnosed in both children.

Three children were borne premature (week 32, 33 and 35) while the remaining children were early-term babies born between weeks 37 and 38. One child was delivered by caesarean section due to maternal reasons. Eight children had neonatal complications with three children having jaundice, three children having respiratory distress and poor feeding, of which one child also had congenital CMV, one child had neonatal liver failure and one child had hydrops. Table 1 show patient characteristics of the nine patients.

### 3.2. Symptoms at Presentation

All but one child had reported neonatal challenges and complications such as respiratory distress, poor feeding abilities, and/or jaundice. The age of first presenting symptoms retrospectively linked to NPC ranged from 0–37 days. The most prevalent presenting symptoms linked to NPC were jaundice (8/9) and hepatosplenomegaly (8/9), followed by ascites (6/9), pulmonary symptoms (5/9) and hypoglycaemia (5/9). Total bilirubin levels were high with a median of 277 (range: 100–477) microgram/L at time of presentation. Elevated alanine aminotransferase (ALT) was reported in 7/9 with median values of 75 (range: 10–164) U/L and furthermore high median aspartate aminotransferase (AST) levels of 325 (range: 66–531) U/L were reported. Prothrombine time (PTT) was prolonged in all patients with 5/9 children with PT above 40 *s* at time of presentation and an overall median PTT of 46.5 (range: 13–110) seconds. Table 1 shows clinical symptoms and biochemical results at time of presentation.

### 3.3. Transplantation and Diagnosis

In total, 8/9 children were transplanted before the diagnosis of NPC and one patient reported LTx 6 years after diagnosis of NPC. Six children had a split graft, one child a full graft and one child a live related donor, while type of graft was not reported in one child. Reports on explant liver histology showed signs of acute and subacute liver damage ranging from massive necrosis to cirrhosis with regenerative nodules, mixed macro and microvascular cirrhosis and focal fatty changes.

The median time between appearance of first symptoms and LTx in the eight children transplanted before diagnosis was 40 days (range: 10–1151), while the median time between LTx and subsequent diagnosis of NPC was 98 days (range: 1–730). Persistent post-LTx splenomegaly and/or deranged transaminases led to further work up including skin biopsy with filipin staining (7/9) and/or bone marrow aspiration (5/9). Bone marrow aspiration only aided diagnosis in one patient, showing large cells with picnotic nuclei and very large foamy cytoplasm, while filipin stainings were highly positive in all seven children.

Re-evaluation of explanted liver in one centre after diagnosis of NPC reported liver histology with hyperplasia of Kupffer cells and vacuolated cytoplasm with lipidic material in keeping with the diagnosis of NPC. The overall median time between appearance of first symptoms and diagnosis of NPC was 158.5 days (range: 13–1448).

Genetic testing was completed in seven patients. In four patients, mutation analyses were performed by direct sequencing of the NPC genes. A diagnosis of NPC was confirmed from cultured fibroblasts with complementation analysis confirming NPC as the gene responsible in two patients. However, the common c.3182 T > C mutation was not detected and further genetic investigation was declined in one patient. Genetic testing revealed mutations in NPC1 that in many cases correlate with a severe biochemical phenotype, such as p.Ile1061Thr, p.Arg1077X, p.Pro887Leu and a deletion mutant in exon 23. A detailed list of the mutations detected is shown in Table 2.

Chitotriosidase levels were reported in five patients and plasma oxysterol was only reported as being used in one patient. In three children chitotriosidase was negative and hence not helpful in the diagnosis (reported as negative results). Two patients had reported elevated chitotriosidase levels (one reported a level of 672 nmol/h/mL and one patient reported elevated level). Eight patients had reported skin biopsies of which seven were positive. One result was not available. These results were received after the initial acute presentation and after LTx in all but one patient.

### 3.4. Follow-Up

All nine children survived beyond early infancy. At time of data collection, seven children are still alive after LTx with a median follow-up time of 9 (range: 5–13) years, while two children died. Both children died due to pulmonary complications at the age of 5 years.

Before their death, both children developed neurological symptoms with presence of seizures, gelastic cataplexy, dystonia, dysphagia and cognitive deficits. In addition, four children are reported to have developed neurological symptoms during follow-up. The most common symptom is developmental delay and cognitive symptoms, which is reported in three of the four children, followed by gelastic cataplexy in two children and seizures in one child.

In total, three children were treated with Miglustat, of which one child had died. One child (age 7 years) is currently without any neurological deficits while the other child (age 13 years) has developed multiple neurological symptoms with delays in all developmental domains, seizures and gelastic cataplexy. Follow-up time and clinical outcome are shown in Table 2.

## 4. Discussion

Niemann-Pick disease type C is a rare autosomal recessive disorder with an estimated incidence of 1:150,000 live births [12]. In the UK, the clinical heterogeneity of the condition has been confirmed in a recent large review of cases, with figures suggesting approximately a third of patients each present in the newborn, juvenile and adolescent/adult periods [5]. Of those presenting neonatally, the predominant clinical phenotype was of chronic liver dysfunction with prolonged jaundice.

Only very few reports have been published concerning LTx in children who was later diagnosed with NPC. One case report concerns a 5-day-old boy with suspected neonatal hemochromatosis who underwent LTx and later developed progressive neurological symptoms [13]. A second paper reports on three cases of LTx in neonatal acute liver failure who were later diagnosed with NPC [14].

The hallmark of the condition is splenomegaly, usually with hepatomegaly. In most children this cholestatic liver disease is clinically stable, being only occasionally complicated by portal hypertension and variceal bleeding or hepatocellular carcinoma [15,16]. The liver disease usually recovers completely and is then followed by development of neurological disease, the age of onset of which is currently unpredictable. The slow progression of the liver disease usually allows time for the accurate diagnosis of the disease for those with onset of symptoms at older age. However, slow progression of liver disease was not seen in the majority of this manuscript’s study cohort, hence why LTx occurred before the diagnosis could be made. The gold standard is the demonstration of intra-lysosomal accumulation of unesterified cholesterol with a characteristic pattern of staining with filipin, together with demonstration of impaired esterification of exogenous LDL-derived cholesterol [9,17]. However this requires testing on fibroblasts and introduces a significant delay into the confirmation of the diagnosis with turnaround times of up to four months depending on the success of culture. Alternatively the detection of two pathogenic mutations in the NPC1 (responsible for >95% of cases) or NPC2 genes can confirm the diagnosis though this is only available in a few centres worldwide and many different mutations have been described to date [18,19]. One mutation in NPC1 (c.3182 T > C) is observed at a higher frequency than others and is found in a significant proportion of neonatal presentations [4,20].

Other markers of storage disorders are recommended as initial screens though they lack sensitivity and specificity. Liver biopsy specimens do not consistently demonstrate storage material, which can be hard to differentiate from activated Kupffer cells in the face of active liver disease [16]. Bone marrow aspiration for storage cells has also been shown recently to fail to diagnose NPC in a significant number of cases, especially in the context of infantile liver disease [21]. The sensitivity of this test could potentially be improved by filipin staining on bone marrow aspirate material; however, the expertise to perform this is not widely available. A raised plasma chitotriosidase level has been reported to be predictive of storage disorders, including NPC, but the levels seen in NPC are lower than those seen in Gaucher disease and case reports demonstrate the test does not pick up all cases [22]. Additionally, the not-infrequently encountered pseudodeficiency state can render it unreliable in some cases. The specificity for NPC of a raised chitotriosidase level alone is low as it may be elevated in other macrophage diseases such as haemophagocytosis. In our cohort, only 2/9 showed a raised level.

ALF is a much rarer neonatal presentation of NPC and in the UK study, severe liver disease was only noted in three patients, of whom two died within the first 8 weeks of life [4,5]. North American data reports this presentation in no more than 10% of all NPC cases (C Hendriksz: Personal Communication). This clinical scenario brings with it a number of diagnostic dilemmas. Firstly the ancillary investigations such as liver biopsy and bone marrow aspiration are unhelpful to demonstrate storage material. Secondly the rapid progression of neonatal liver failure gives only a short time window in which to complete further investigation before LTx becomes necessary and this time frame is not compatible with diagnostic tests involving fibroblast culture.

It has been recognised that LTx does not protect against neurological deterioration in NPC, which is confirmed by our case series. Birch et al. also demonstrate that recurrent disease in the transplanted liver is common. Given the paucity of paediatric liver donors, especially for the neonatal age group, LTx in the context of a neurodegenerative condition for which there is as yet no cure would appear to be a suboptimal use of a precious resource, although this has been considered in view of the unpredictable neurological progression of the condition and wide clinical spectrum [15].

In conclusion, these nine cases demonstrate that early diagnosis of NPC continues to be a challenge in acute neonatal liver failure, and the diagnosis may only be reached after LTx, even where suspicion of a storage disorder may be raised (for example from the presence of splenomegaly). Therefore, follow-up with genetic testing and keeping an open mind to etiologic diagnosis is recommended. It would seem that, due to the difficulties discussed, this may be unavoidable, although advances in genomic DNA microarray technology may eventually make screening for the more common NPC1 mutations more practical within this timescale. Furthermore, neurological disease including gaze anomalies, hearing loss, unsteady gait, seizures, cognitive impairment and developmental delay are not prevented in the majority of patients but symptom-free survival has been observed between 6–9 years in 1/3 of the of cases. Genotype does not appear to predict neurological outcome after transplantation and LTx remains controversial in NPC. 

## Figures and Tables

**Table 1 children-08-00819-t001:** Patient characteristics and clinical presentation of nine patients diagnosed with Niemann Pick type C who underwent liver transplantation between 2003 and 2018.

Demographic Variable	Result	*n*
Male	6 (67)	9
**Family history**		
Consanguinity	1 (11)	9
Death among siblings	1 (11)	9
Miscarriages and termination	2 ^1^ (22)	9
**Birth**		
Gestational age week, median (range)	37 (32–38)	8 ^2^
Birth weight, kg, median (range)	2.9 (2.4–3.5)	8 ^2^
Caesarean section	1 ^3^ (11)	9
Fetal hydrops	1 (11)	9
Fetal ascites	1 ^4^ (11)	9
Age at first clinical symptom, days, median (range)	7 (0–37)	9
**Clinical presentation**		
Jaundice	8 (89)	9
Hepatomegaly	8 (89)	9
Splenomegaly	8 (89)	9
Ascites	6 (67)	9
Respiratory distress	5 (56)	9
Hypoglycaemia	4 (44)	9
Other	2 ^5^ (22)	9
**Biochemistry at presentation, median (range)**		
Total bilirubin, micromol/L	277 (100–477)	8
Alanine aminotransferase, U/L	75 (10–164)	8
Aspartate aminotransferase, U/L	325 (66–531)	8
Gamma glutamyl transferase, U/L	52 (15–330)	8
Prothrombine time, seconds	46.5 (13–110)	8
PT > 40 s	5 (56)	9
Albumin, g/L	27 (13–39)	8
Ammonia, µmol/L	74 (55–176)	6
Lactate, mmol/L	2 (2–3)	4

Data in the table are presented as n (%), unless otherwise specified. CMV: cytomegalovirus, LTx: Liver transplantation, NPC: Niemann Pick type C. ^1^ One woman had two terminations (first termination because of missing heartbeat at 3 months, second termination due to NPC diagnosed in utero) another reported sporadic miscarriages and one infant death not characterised further, ^2^ One missing, ^3^ Maternal indication, ^4^ The child was also reported to present with transient respiratory distress and congenital CMV infection, ^5^ One child had impaired neonatal neurological examination and one child had fluid retention in lower limbs.

**Table 2 children-08-00819-t002:** Mutations detected in 9 patients diagnosed with Niemann Pick type C and the clinical outcome after liver transplantation.

Case	Country	Sex	NPC1 Mutations	Outcome	Follow-Up Time
1	Canada	M	NA ^1^	No reported issues	7 years
2	Canada	M	c.3229C > T and c.3182 T > C	Developmental delay, seizures and gelastic cataplexy	13 years
3	Germany	F	NA ^1^	Gelastic cataplexy	NA
4	UK	M	Heterozygeus for missense variant of NPC1 gene: Phe1207Ser in exon 24; c.3591 + 4delA in exon 23	No reported issues	8 years
5	UK	F	NA ^1^	DEAD ^2^	5 years
6	Germany	M	NA ^1^	Cognitive impairment and developmental delay	13 years
7	Germany	M	Compound Heterozygote in the NPC1 gene: p.Ile1061Thr het; p.(Pro887Leu) het.	Cognitive impairment and developmental delay	6 years
8	Australia	F	p.I1061T homozygote	No reported issues	9 years
9	Spain	M	p.T1066N/p.T1066N	DEAD ^3^	NA

LTx: Liver transplantation, NPC: Niemann-Pick disease type C. ^1^ No mutation analysis, ^2^ NPC related death, ^3^ Respiratory failure following pneumonia.

## Data Availability

The datasets generated and analysed during the current study are not publicly available due to privacy and ethical concerns in light of the rare nature of the reported disease but are available from the corresponding author on reasonable request.

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
