# Peer review of "A Case Series on Genotype and Outcome of Liver Transplantation in Children with Niemann-Pick Disease Type C"

_children, 2021, doi:10.3390/children8090819_

Round 1
Reviewer 1 Report
This paper is written about outcome of liver transplantation(LTx) in 9 patients with NPC. The authors concluded that neurological symptoms are not prevented in the majority of patients and genotype does not appear to predict neurological outcome after LTx. This is a valuable report for clinicians, however, there are several problems.
- It is well known that neonatal presentation of liver disease and the onset and progression of neurological disease is not correlated in NPC. In order to show the effect of LTx, it would be important to compare the onset age of neurological symptoms between the same genotype patients with and without LTx. The authors should add the onset age of neurological symptoms in each patients and discussed about it.
- Most patients were undergone LTx before the diagnosis of NPC. Recently, several biomarkers are proposed for NPC. The authors should discuss about possibility of early diagnosis using these biomarkers.
- Table 2 is hard to read. The author should add transverse lines between the cases.
- ‘NPC2’ should be replaced correctly in p.2-4.
Author Response
Reviewer 1:
1. It is well known that neonatal presentation of liver disease and the onset and progression of neurological disease is not correlated in NPC. In order to show the effect of LTx, it would be important to compare the onset age of neurological symptoms between the same genotype patients with and without LTx. The authors should add the onset age of neurological symptoms in each patients and discussed about it.
This is a very relevant comment which we agree would add to the understanding of the disease process. Unfortunately the ages of onset of neurological symptoms are not as reliable as the dates reported on diagnosis. We have collected the most reliable dates during the data collection process which has been challenging considering the time period covered and the multiple centers involved. Despite this lack of precise age at onset of neurological deterioration, we believe that the manuscript add to the current sparse pool of reports about NPC and the clinical course after LTX.
2. Most patients were undergone LTx before the diagnosis of NPC. Recently, several biomarkers are proposed for NPC. The authors should discuss about possibility of early diagnosis using these biomarkers.
Thank you for pointing out this relevant issue. There are new guidelines on investigation of NPC and although it is still not fool proof this is much easier than before when using a “cholestasis gene panel” and a combination of biomarkers such as oxysterols (which indeed are not ideal for patients with cholestasis), NPC bile acids and SM509 ratio. However, these possibilities were not available for the patients involved in this case series. The issue has been addressed in the paper.
3. Table 2 is hard to read. The author should add transverse lines between the cases.
We have formatted the tables according to the journal standard. We are happy to provide transverse lines if it’s accepted by the journal.
4. ‘NPC2’ should be replaced correctly in p.2-4.
Thank you for noticing this type error, it has been corrected.
Reviewer 2 Report
Thank you for submitting a manuscript of "Poor correlation between genotype and outcome of liver transplantation in children with Niemann-Pick Disease type C".
This is a very interesting topic. However, there are several concerns.
- The authors suggested that genotype does not appear to predict neurological outcome after LTx. However, the number of subjects seems to be insufficient to discuss the correlation. I recommend that this paper be submitted as case series.
- Would you explain the possible mechanism of the prolonged neurological symptom free survival?
Author Response
Reviewer 2:
1. The authors suggested that genotype does not appear to predict neurological outcome after LTx. However, the number of subjects seems to be insufficient to discuss the correlation. I recommend that this paper be submitted as case series.
Thank you for this remark. We completely agree. This is a case series and because of the rare nature of the disease and the even more rare incidents of LTX in these patients it’s impossible to make statistical sound correlation analysis. We have changed the title of the manuscript to “Case series report on genotype and outcome of liver transplantation in children with Niemann-Pick Disease type C”
2. Would you explain the possible mechanism of the prolonged neurological symptom free survival?
Thank you for this very relevant question. We would be overstepping our knowledge based on our research results trying to answer this question. It’s immensely relevant in a clinical setting and we wish we could explain and use rock solid validated data to support our daily practice in the rare cases where we meet families who find themselves in the stressful situation of having to grasp a NPC diagnosis.
Reviewer 3 Report
This paper is a retrospective review of a controversial indication for liver transplantation in children. Clinical interest is based on the importance of liver transplantation for children to survive in acute liver failure. The series of 9 patients is small although it confirms that neurological lesions can progress. The methodology and discussion are correct and the bibliography adequate.
Author Response
Reviewer 3:
This paper is a retrospective review of a controversial indication for liver transplantation in children. Clinical interest is based on the importance of liver transplantation for children to survive in acute liver failure. The series of 9 patients is small although it confirms that neurological lesions can progress. The methodology and discussion are correct and the bibliography adequate.
Thank you very much for the kind comments. We are so happy to hear that you acknowledge our work and the patients and families’ contribution to increase the knowledge of this rare disease. Many thanks.
Round 2
Reviewer 2 Report
Thank you for giving me the opportunity to review the manuscript. The authors emphasized the difficult situation of NPC diagnosis. The authors tried hard to answer reviewers’ previous comments. However, there are still several concerns.
- The authors mentioned that "prolonged" symptom free survival has been observed between 6-9 years in 1/3 of the of cases in discussion section. However, it does not seem that the data related to this is well presented in result section.
- And if this is true, do the authors think liver transplantation helps prognosis of NPC patients? Because liver transplantation has increased the asymptomatic period?
Author Response
Point-by-point response:
- 1. The authors mentioned that "prolonged" symptom free survival has
been observed between 6-9 years in 1/3 of the of cases in discussion
section. However, it does not seem that the data related to this is well
presented in result section.
Thank you for the additional comment on the survival period of the transplanted NPC patients. We agree that the word “prolonged” can be misunderstood and related to the LTx itself. We have no basis to say this based on this small case series. What we report is observations and we hoped to link genetics to the outcome, but unfortunately, this was not the case. We have deleted “prolonged” in the manuscript to avoid confusion when interpreting the reported data. The observed symptom free period can be used as a basic knowledge for clinicians and families who find themselves in this rare situation.
We have collected as many data as possible about the follow-up period and the clinical outcome but the data is limited due to the retrospective nature of the data collection (no possibility to go back and re-collect more data on site).
- 2. And if this is true, do the authors think liver transplantation
helps prognosis of NPC patients? Because liver transplantation has
increased the asymptomatic period?
Because of the nature of the data, we can’t say anything about cause and effect. LTx only avoided immediate death by ALF but didn’t prevent disease deterioration and unfortunately we couldn’t find any genetic clues to predict outcome of the disease.
Kind regards,
Line Modin
